# Analysis of Challenges Due to Changes in Net Load Curve in South Korea by Integrating DERs

**Chi-Yeon Kim [1], Chae-Rin Kim [1], Dong-Keun Kim [2] and Soo-Hwan Cho [1,\*]** 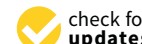

[1]  Department of Electrical Engineering, Sangmyung University, Seoul 03016, Korea;
   201931089@sangmyung.kr (C.-Y.K.); happy980320@naver.com (C.-R.K.)
[2]  Department of Intelligent Engineering Information for Human, Sangmyung University, Seoul 03016, Korea;
   dkim@smu.ac.kr
\*  Correspondence: shcho@smu.ac.kr

**Abstract:** The development of Distributed Energy Resources (DERs) is essential in accordance with the mandatory greenhouse gas (GHG) emission reduction policies, resulting in many DERs being integrated into the power system. Currently, South Korea is also focusing on increasing the penetration of renewable energy sources (RES) and EV by 2030 to reduce GHGs. However, indiscriminate DER development can give a negative impact on the operation of existing power systems. The existing power system operation is optimized for the hourly net load pattern, but the integration of DERs changes it. In addition, since ToU (Time-of-Use) tariff and Demand Response (DR) programs are very sensitive to changes in the net load curve, it is essential to predict the hourly net load pattern accurately for the modification of pricing and demand response programs in the future. However, a long-term demand forecast in South Korea provides only the total amount of annual load (TWh) and the expected peak load level (GW) in summer and winter seasons until 2030. In this study, we use the annual photovoltaic (PV) installed capacity, PV generation, and the number of EV based on the target values for 2030 in South Korea to predict the change in hourly net load curve by year and season. In addition, to predict the EV charging load curve based on Monte Carlo simulation, the EV users' charging method, charging start time, and State-of-Charge (SoC) were considered. Finally, we analyze the change in hourly net load curve due to the integration of PV and EV to determine the amplification of the duck curve and peak load time by year and season, and present the risks caused by indiscriminate DERs development.

**Keywords:** net load curve; PV generation; Monte Carlo simulation; EV charging load prediction; ToU (Time-of-Use) tariff

## 1. Introduction

With the adoption of the 2015 Paris Agreement, which is the basis for the new climate regime, interest in reducing GHG emissions is increasing worldwide. Accordingly, many countries are striving to reduce the use of fossil fuels, a major cause of GHGs, by increasing RES such as PV and wind power [1–3]. In addition, 16% of man-made carbon dioxide ($CO_2$) is produced by motor vehicles (cars, trucks, and buses); therefore, increasing the number of EV is essential to reducing GHGs [4]. According to the "IEA's Global EV Outlook" announced in June 2019, approximately 2 million electric vehicles were sold worldwide in 2018, and the cumulative supply exceeded 0.5 million units. In China, approximately 1.1 million EVs were sold in 2018 and a total of 2.3 million EVs were distributed, accounting for approximately 45% of the EVs worldwide. In addition, 1.2 million EVs in Europe and 1.1 million EVs in the United States were distributed by 2018 [5].

In response to these changes, South Korea is also implementing policies on the supply of RES and EVs. First, for the energy sector, which accounts for more than two-thirds of the GHGs, South Korea announced the "Renewable Energy 3020 Implementation Plan" in December 2017 to reduce GHGs [6]. According to this policy, the installed capacity of PV and wind power in South Korea will be increased from 5.7 GW to 36.0 GW and from 1.2 GW to 17.7 GW, respectively, until 2030 to meet 20% of the total energy generation with renewable energy generation. Second, for the transportation sector, the South Korean government will provide various support services such as constructing public charging infrastructures and providing subsidies for EV purchases and charger installations to increase the penetration of EV [7]. Additionally, according to the "Future Automobile Industry Development Strategy" published by the Ministry of Trade, Industry and Energy of South Korea, the number of EV in South Korea is expected to continue to grow, reaching an accumulated 3 million units by 2030 [8].

Despite the contributions that DERs such as PV, wind power, and EVs make to reduce GHGs, the increase in DERs has negative effects on the load curve. In particular, the load curve decreases during the daytime due to the increase in PV installed capacity and PV generation, resulting in a duck curve. The duck curve is a phenomenon in which the PV generation during the day decreases from the existing camel-shaped load curve to become a duck-shaped curve [9–11]. Previous studies address various problems in power systems such as ramping events, peak load time shifts due to duck curves and irregular generation depending on weather condition [12,13]. In addition, reference [14] says that the increasing PV penetration ratio and generation can cause a net load demand around the afternoon time to significantly decrease, resulting in an increase in the amount of attenuation of thermal power generators, which in turn leads to economic losses. Figure 1 shows a difference of actual net load curves in South Korea on November 20 and November 21 in 2018. According to the Korea Meteorological Administration (KMA) information, it was clear on November 20 and cloudy on November 21 [15]. Table 1 shows that total amounts of energy consumption were 1,525,173 GWh on November 20 and 1,557,242 GWh on November 21, which means that the difference of energy consumption was 32,069 GWh in just one day because of the change of weather condition. When we consider the additional EV charging effect, it will make the duck curve worse, especially after 8 PM when the need for EV charging is intensive, by making a large difference between the peak load and the off-peak load.

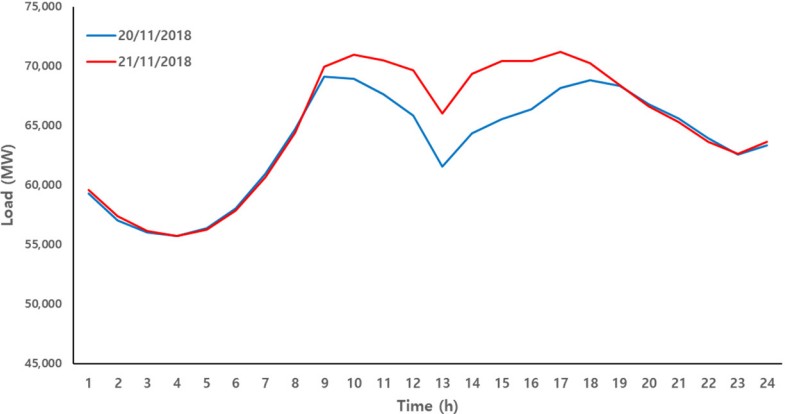

**Figure 1.** Net load curves for 20/11/2018 and 21/11/2018.

**Table 1.** Net load difference between 20/11/2018 and 21/11/2018.

| Date | 20/11/2018 | 21/11/2018 |
|---|---|---|
| Net Load (GWh) | 1,525,173 | 1,557,242 |
| Difference (GWh) | 32,069 | |

Other previous work has mainly dealt with overall negative impacts of the duck curve, and long-term power demand forecasting such as the "8th Basic Plan for Long-term Electricity Supply and Demand" [16] includes the total amount of annual load (TWh) and the expected peak load level (GW) in summer and winter seasons until 2030. However, in order to accurately analyze the change of net load curve pattern, it is necessary to predict the hourly pure load curve and the hourly net load curve which includes the pure load level, demand response, renewable generation and other factors. Some studies have introduced the hourly net load pattern with renewable energy resources, but these were based on only the current load level and mainly focused on the impact of variability of renewables [12,17]. Now, we need for the hourly load pattern prediction considering the future change of energy circumstances until 2030. Thus, we predict the hourly load curve and the hourly net load by considering PV installed capacity, PV generation, and the number of EVs based on the target of South Korean policy until 2030.

Moreover, the current time ranges of ToU tariff and demand response (DR) in South Korea will be not suitable for the future hourly net load pattern. Table 2 shows the current seasonal time range for ToU tariff in South Korea [18]. The peak load time will be shifted into other time ranges as shown in Figure 1. Additionally, DR is implemented from 9 AM to 8 PM (except 12 PM~1 PM) to reduce electricity demand in South Korea [19]. Thus, as the number of EVs charging after 8 PM increases, it is necessary to change the DR operating time to manage new peak loads occurring after 8 PM due to EV charging load.

**Table 2.** ToU tariff in South Korea [18].

| Season | Spring/Fall | Summer | Winter |
|---|---|---|---|
| Light Load | 23:00~09:00 | 23:00~09:00 | 23:00~09:00 |
| Medium Load | 09:00~10:00<br>12:00~13:00<br>17:00~23:00 | 09:00~10:00<br>12:00~13:00<br>17:00~23:00 | 09:00~10:00<br>12:00~17:00<br>20:00~22:00 |
| Peak Load | 10:00~12:00<br>13:00~17:00 | 10:00~12:00<br>13:00~17:00 | 10:00~12:00<br>17:00~20:00<br>22:00~23:00 |

In this paper, Section 2 predicts the load curve to determine the change of the net load curve and the peak load time by year and season according to the increase in penetration of PV in South Korea. Section 3 predicts the EV charging load curve based on Monte Carlo simulation taking the increase in the number of EV into account. Section 4 takes the results of Sections 2 and 3 into account to predict the net load curve and peak load time by year and season caused by the increase in penetration of PV and EV, and presents the need to change existing operation system such as ToU tariff and DR in South Korea.

## 2. Changes in Net Load Curve According to Increased PV Penetration

In the past, when the PV installed capacity and PV generation were not large, the ToU tariff and DR operation to manage electricity demand were very effective. Figure 2 shows the seasonal average net load curve for South Korea in each year from 2015 to 2017 based on actual data obtained from Korea Electric Power Exchange (KPX), which is responsible for market and system operators [20]. The seasonal peak load time in each year from 2015 to 2017 in Table 3.

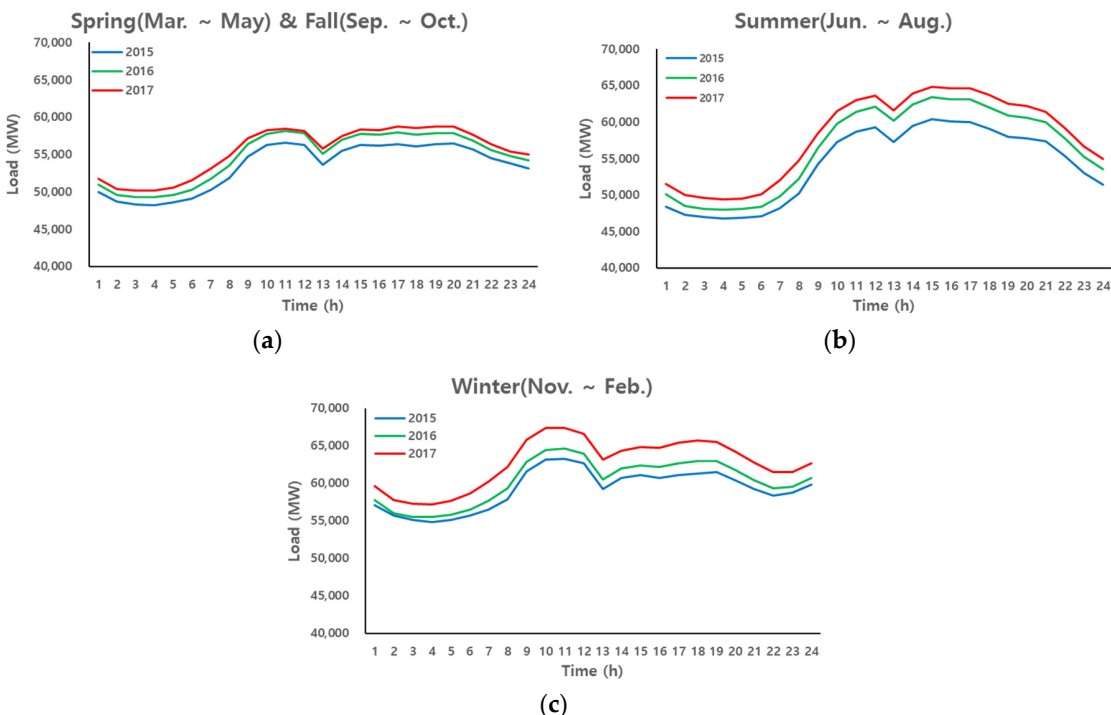

**Figure 2.** Seasonal average net load curves for South Korea in each year from 2015 to 2017: (**a**) for spring (March, April, May) and fall (September, October); (**b**) for summer (June, July, August); and (**c**) for winter (November, December, January, February).

**Table 3.** Seasonal peak load time in each year from 2015 to 2017 in Figure 2.

| Season | 2015 | 2016 | 2017 |
|---|---|---|---|
| Spring/Fall | 11 AM | 11 AM | 5 PM |
| Summer | 3 PM | 3 PM | 3 PM |
| Winter | 11 AM | 11 AM | 11 AM |

The seasonal peak load time in each year from 2015 to 2017 is suitable for ToU tariff and DR operating time shown in Table 3. However, the peak load time for the spring/fall changed from 11 AM in 2015 and 2016 to 5 PM in 2017, unlike for the summer and winter. In other words, due to the constant increase in PV installed capacity and PV generation, a duck curve occurs and the peak load time shifts from 11 AM to 5 PM. This fact means that cooling and heating loads are not frequently used in spring and fall unlike summer and winter, so the peak load time due to increasing PV generation is shifted earlier than other seasons. In this section, to determine the shift in the peak load time by year and season, the load curve, the net load curve, PV installed capacity, and PV generation are predicted according to South Korea's energy related policies.

## 2.1. Introduction to Prediction Method of The Future Net Load Curve

A net load curve (or pattern) in a day is obtained by removing a PV generation curve from a load curve [21]. Therefore, to determine the future net load curve, it is necessary to predict the future load curve and the future PV generation curve. The future load curve is derived by combining the current load curve with the values of Table 4, which are the estimated future load growth rates considering GDP, electricity price, population, temperature and so on [16]. In our study, we additionally consider the expected EV charging load curve obtained by Monte Carlo Simulation. The future PV generation curve is derived by combining the seasonal PV generation curve and the expected PV installed capacity. A process to predict a net load curve of 2018 is summarized as follows:

(1)　To create the load curve of 2017, we combined seasonal PV generation curves of 2017 shown in Figure 3 with a net load curve for 2017 provided by KPX. The seasonal PV generation curves are based on the actual data measured and used by KPX. In this case, we ignored the EV charging load curve because few EV were available in South Korea in 2017. A detailed prediction method for the future EV load curve will be introduced at the Section 3.

(2)　To predict the load curve of 2018, we scaled up the load curve of 2017 obtained in step (1) by 2.85% in Table 4.

(3)　To predict the seasonal PV generation curves of 2018, we scaled up the seasonal PV generation curves by considering the PV generation information of 2018 in Table 5. Table 5 shows the actual PV installed capacity and PV generation for 2016, 2017, and 2018 obtained from KPX, Statistics South Korea, and the Korea Electric Power Corp. This step will be addressed at the next Section 2.2 in details.

(4)　To determine the predicted net load curve of 2018, we removed the PV generation curve obtained in step (3) from the load curve of 2018 obtained in step (2).

(5)　Finally, to verify the predicted net load curve of 2018, we compared it with the actual net load curve of 2018 provided by KPX. The results are shown in Figure 4 and Table 6. The seasonal error rate between the actual net load and the predicted net load for 2018 is 0.453% in spring/fall, 0.740% in summer, 0.846% in winter, and an average annual error is 0.631%, indicating a high prediction accuracy. Now, we expand this process to estimate the long-term load curves until 2030.

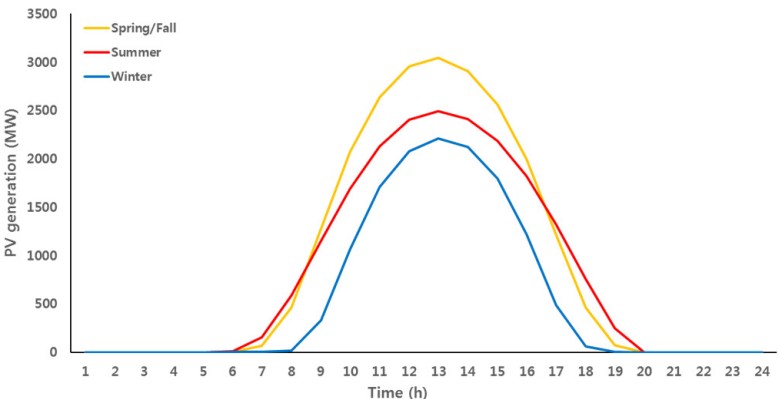

**Figure 3.** Seasonal PV generation curves of 2017.

**Table 4.** Load growth rate from 2018 to 2030 [16].

| Year | 2018 | 2019 | 2020 | 2021 | 2022 | 2023 | 2024 |
|------|------|------|------|------|------|------|------|
| **Rate (%)** | 2.85 | 2.76 | 2.66 | 2.61 | 2.28 | 2.16 | 2.01 |
| **Year** | 2025 | 2026 | 2027 | 2028 | 2029 | 2030 | - |
| **Rate (%)** | 1.94 | 1.83 | 1.72 | 1.58 | 1.51 | 1.40 | - |

**Table 5.** PV installed capacity, PV generation, and generation time for 2016, 2017, and 2018.

| Year | 2016 | 2017 | 2018 |
|------|------|------|------|
| **PV Installed Capacity (GW)** | 4.5 | 5.83 | 8.10 |
| **PV Generation (TWh)** | 5.122 | 7.056 | 9.208 |
| **Average Daylight Time (Hour/day)** | 3.11 | 3.32 | 3.11 |

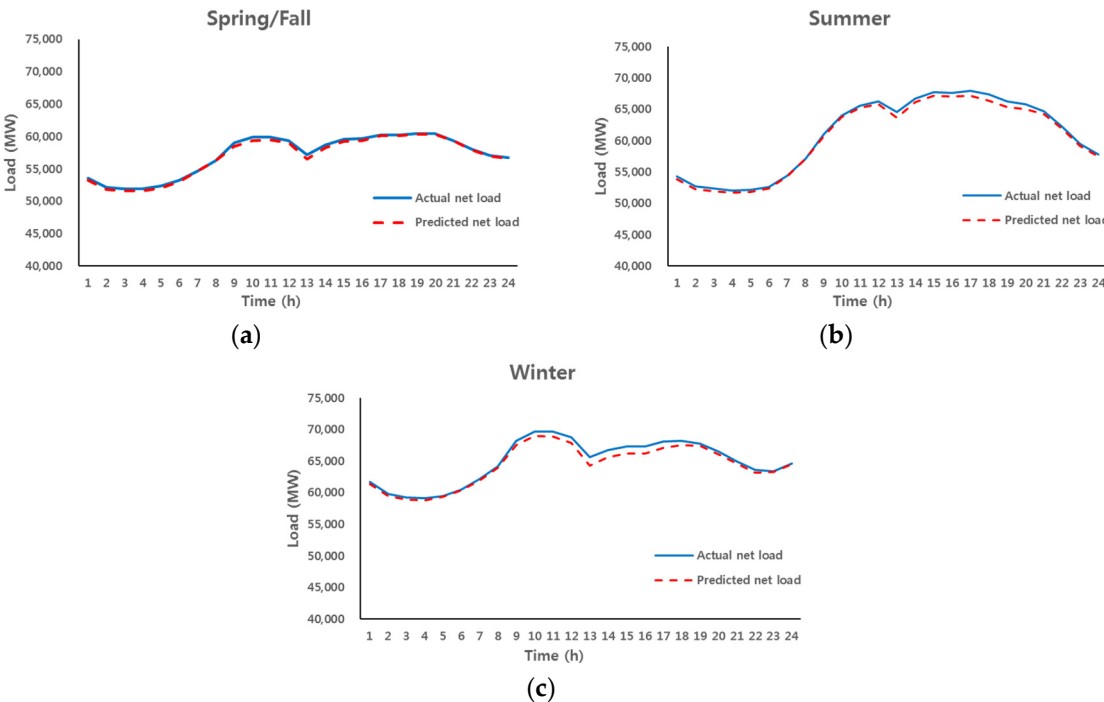

**Figure 4.** The comparison of the actual net load curve for 2018 and the predicted net load curve by season for 2018: (**a**) The actual net load curve vs. predicted net load curve in spring/fall; (**b**) The actual net load curve vs. predicted net load curve in summer; (**c**) The actual net load curve vs. predicted net load curve in winter.

**Table 6.** Seasonal error rate between the actual net load and predicted net load for 2018.

| Season | Spring/Fall | Summer | Winter | Total |
|---|---|---|---|---|
| Actual net load (TWh) | 251.111 | 134.607 | 140.119 | 525.838 |
| Predicted net load (TWh) | 249.971 | 133.612 | 138.934 | 522.517 |
| Error rate (%) | 0.453 | 0.740 | 0.846 | 0.631 |

### 2.2. Prediction of PV Installed Capacities and PV Generations Until 2030

To predict load curves and net load curves until 2030, PV installed capacity data and PV generation curves until 2030 are also needed. According to the "Renewable Energy 3020 Implementation Plan" [6], South Korea will expand the PV installed capacity of 5.83 GW in 2017 to 36.5 GW until 2030 in order to supply 20% of the energy generation with renewable energy generation. In addition, the report titled "Korea Energy Vision 2050" indicates that the RES installed capacity and the PV installed capacity must reach 41.0 GW and 23.0 GW, respectively, by 2025 to meet the objectives of the plan [22]. Table 7 shows the targets for the RES installed capacity, PV installed capacity, and PV generation in accordance with the above-mentioned national plans and reports. Moreover, according to the report, to meet the renewable energy generation target for 2030, the proportion of energy generation by PV must account for 33% of the total renewable energy generation. In other words, about 6.6% of the total annual energy generation in South Korea in 2030, corresponding to 44.846 TWh, should be supplied by PV, as shown in Table 7. By using the second-order approximation method with some target values in Table 7, we estimated PV installed capacity and PV generation for each year from 2019 to 2030, as shown in Table 8.

**Table 7.** Summary of RES, PV installed capacity and PV generation to meet the plans [22].

| Year | 2017 | 2025 | 2030 |
|---|---|---|---|
| **RES Installed Capacity (GW)** | 15.1 | 41.0 | 63.8 |
| **PV Installed Capacity (GW)** | 5.83 | 23.0 | 36.5 |
| **PV Generation (TWh)** | 7.056 | - | 44.846 |

**Table 8.** The predicted PV installed capacity and PV generation from 2019 to 2030.

| Year | 2019 | 2020 | 2021 | 2022 | 2023 | 2024 |
|---|---|---|---|---|---|---|
| **PV Installed Capacity (GW)** | 9.84 | 11.84 | 13.92 | 16.09 | 18.34 | 20.68 |
| **PV Generation (TWh)** | 11.37 | 13.71 | 16.16 | 18.74 | 21.44 | 24.27 |
| **Year** | **2025** | **2026** | **2027** | **2028** | **2029** | **2030** |
| **PV Installed Capacity (GW)** | 23 | 25.61 | 28.20 | 30.88 | 33.64 | 36.50 |
| **PV Generation (TWh)** | 27.21 | 30.28 | 33.47 | 36.78 | 40.21 | 44.84 |

### 2.3. The Predicted Results of Load Curves, PV Generation Curves and Net Load Curves Until 2030

The predicted results of the seasonal load curve are shown in Figure 5. For better legibility, only even-numbered years from 2018 to 2030 and a reference year of 2017 are shown in the load curves. In addition, the values of predicted load for each year are shown in Table 9.

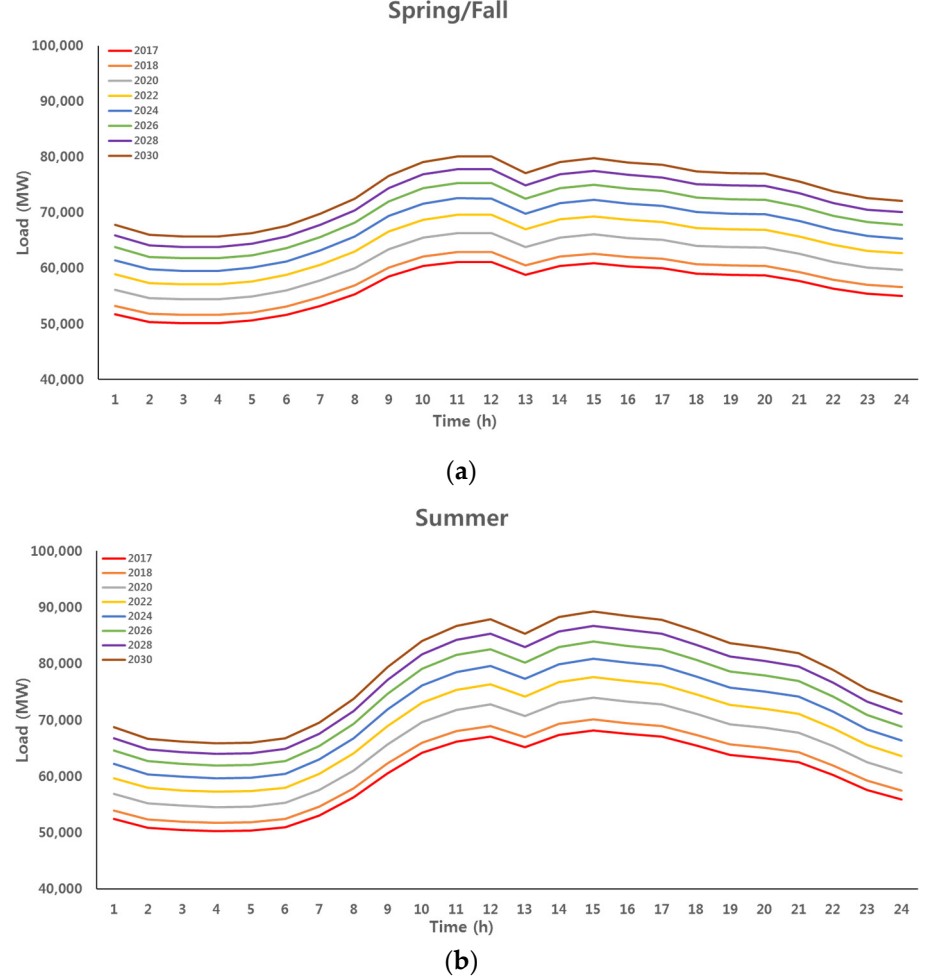

**Figure 5.** *Cont.*

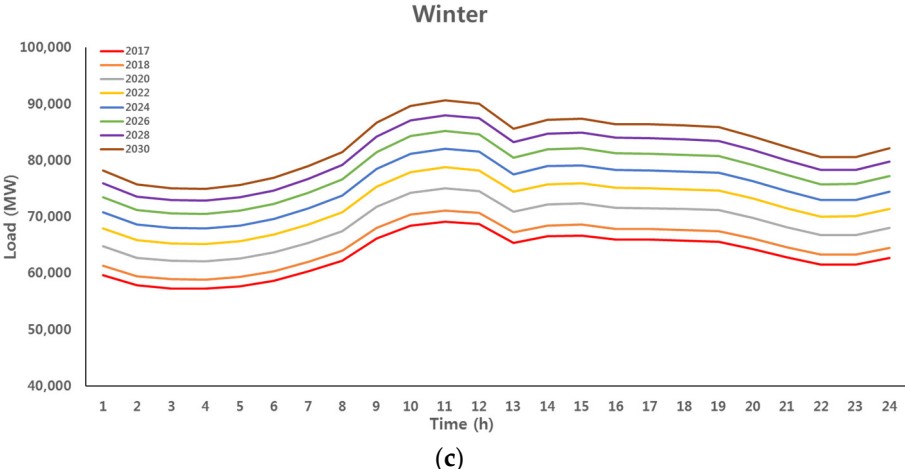

**Figure 5.** Seasonal predicted load curves in each year from 2017 to 2030: (**a**) for spring and fall; (**b**) for summer; and (**c**) for winter.

**Table 9.** The predicted load from 2017 to 2030.

| Year | 2017 | 2018 | 2019 | 2020 | 2021 | 2022 | 2023 |
|---|---|---|---|---|---|---|---|
| **Predicted Load (TWh)** | 514.80 | 529.48 | 544.09 | 558.57 | 573.14 | 586.19 | 598.86 |
| **Year** | 2024 | 2025 | 2026 | 2027 | 2028 | 2029 | 2030 |
| **Predicted Load (TWh)** | 610.92 | 622.78 | 634.18 | 645.11 | 655.30 | 665.19 | 674.52 |

And to predict PV generation curves until 2030, we used the actual PV generation curve of 2017 shown in Figure 3 and the predicted PV generation until 2030 in Table 8. Figure 6 shows the predicted PV generation curve from 2017 to 2030 in spring/fall.

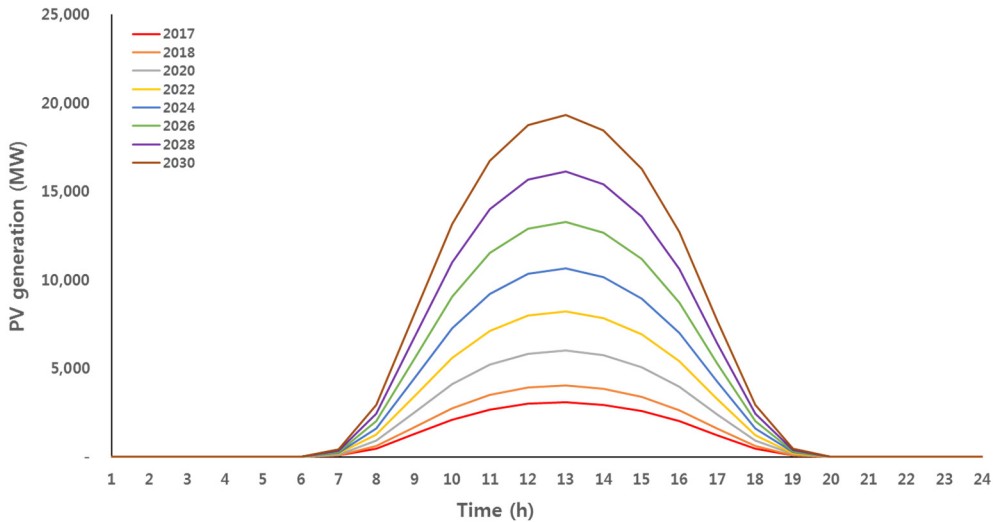

**Figure 6.** The predicted PV generation curves from 2017 to 2030 in spring and fall.

The predicted results of the seasonal net load curve are shown in Figure 7. The predicted net load curves until 2030 are obtained by removing the predicted PV generation curves in Figure 6 from the predicted load curves in Figure 5. The predicted net load curves are displayed at 1-year intervals from 2015 to 2017, and the predicted net load curves from 2018 to 2030 are displayed at 2-year intervals. Figure 7 shows that the actual net load curve from 2015 to 2017 increases steadily, and the predicted

net load curve from 2018 to 2030 also increases in a similar pattern. The predicted net load is shown in Table 10, and the seasonal peak load time with the increase in penetration of PV is shown in Table 11.

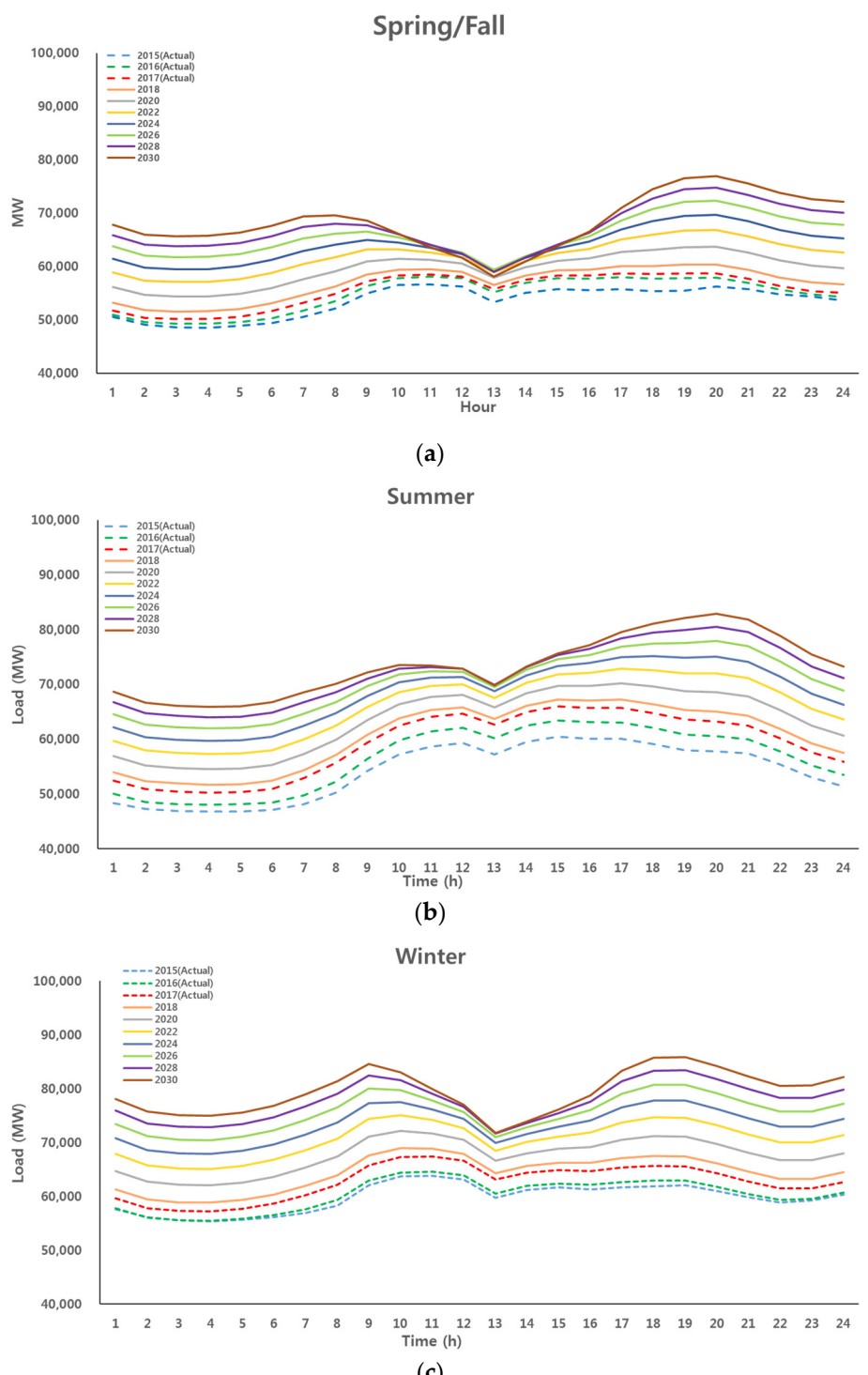

**Figure 7.** Seasonal predicted net load curves in each year from 2017 to 2030 considering only PV: (**a**) for spring and fall; (**b**) for summer; and (**c**) for winter.

**Table 10.** The predicted net load from 2018 to 2030.

| Year | 2015 | 2016 | 2017 | 2018 | 2019 | 2020 | 2021 | 2022 |
|---|---|---|---|---|---|---|---|---|
| **Predicted Net Load (TWh)** | 483.46 (Actual) | 497.02 (Actual) | 507.75 (Actual) | 522.52 | 535.06 | 547.29 | 559.51 | 570.08 |
| **Year** | 2023 | 2024 | 2025 | 2026 | 2027 | 2028 | 2028 | 2030 |
| **Predicted Net Load (TWh)** | 580.14 | 589.47 | 598.48 | 606.91 | 614.75 | 621.72 | 628.28 | 633.88 |

**Table 11.** Peak load time by season from 2015 to 2030 considering only PV penetration.

| Year | Spring/Fall | Summer | Winter |
|---|---|---|---|
| 2015 | 11 AM | 3 PM | 11 AM |
| 2016 | 11 AM | 3 PM | 11 AM |
| 2017 | 5 PM | 3 PM | 11 AM |
| 2018 | 8 PM | 3 PM | 10 AM |
| 2019 | 8 PM | 5 PM | 10 AM |
| 2020 | 8 PM | 5 PM | 10 AM |
| 2021 | 8 PM | 5 PM | 10 AM |
| 2022 | 8 PM | 5 PM | 10 AM |
| 2023 | 8 PM | 5 PM | 10 AM |
| 2024 | 8 PM | 6 PM | 6 PM |
| 2025 | 8 PM | 8 PM | 7 PM |
| 2026 | 8 PM | 8 PM | 7 PM |
| 2027 | 8 PM | 8 PM | 7 PM |
| 2028 | 8 PM | 8 PM | 7 PM |
| 2029 | 8 PM | 8 PM | 7 PM |
| 2030 | 8 PM | 8 PM | 7 PM |

Heating and cooling loads are not used for the spring/fall season; thus, there is already a shift in the peak load time in the net load curve to 8 PM in 2018, which is not suitable for ToU tariff application and DR operating time. In summer, the peak load time in 2019 is shifted from 3 PM to 5 PM, but this is suitable for ToU tariff and DR operating time. However, the peak load time shifts to 6 PM in 2024, which is not suitable for ToU tariff application. In addition, the peak load time shifts to 8 PM in 2025, and the DR operating time is also no longer suitable. For winter, the peak load time changes a total of three times from 2017 to 2030, but remains suitable for the application of the ToU tariff and DR operating time.

## 3. Changes in Net Load Curve According to Increased EV Penetration

The increase in the number of EVs, which are usually charged after work hours, augments the duck curve phenomena and promotes the shift in the peak load time. To determine the effect of these changes, the EV charging load curve is predicted by considering the cumulative number of EVs per year based on Monte Carlo Simulation. The data used to predict EV charging load is probability data based on surveys of actual EV users. Monte Carlo Simulation is the most suitable method to use probability data as a method of performing a simulation by repeating random selection of input variables from a probability distribution [23,24].

In this paper, the following four conditions are assumed to predict the EV charging load curve:

(1)   Charging time is set to 6 PM, after working hours, to analyze the incompatibility of DR operation time and application of ToU tariff by shift of peak load time due to EV charging load curve. According to the report "A Study on the Activation of Electric Vehicle Supply through Survey and Analysis of Actual Purchaser Use", published by the Ministry of Environment of South Korea, the proportion of EVs charged during this period is the largest with 84 % of all EVs [25].

(2)   It is assumed that all EVs are fully charged when charging.

(3)　According to a report in [25], 86.2% of all EV users use personal chargers. Therefore, only personal chargers are considered for the charging method to predict the EV charging load.

(4)　The parameters for the EVs used in the simulation to predict charging loads are shown in Table 12, which refer to the "2017 Electric Vehicle Survey Results Report" released by the Ministry of Trade, Industry, and Energy of South Korea [26]. According to this report, the average daily driving distance of EVs is 48 km. The daily consumption capacity of each EV is calculated by the average daily driving distance and fuel economy. Additionally, the daily charging capacity is the same as the daily consumption capacity because it is always assumed to be fully charged.

**Table 12.** The parameters for the EVs used in the simulation to predict charging load curve.

| EV Models | Ionic | Soul | SM3 |
|---|---|---|---|
| Avg. driving distance (km/day) | 48 | 48 | 48 |
| Ratio (%) | 57 | 27 | 16 |
| Battery capacity (kWh) | 28 | 27 | 33 |
| Fuel efficiency (km/kWh) | 6.3 | 5 | 4.4 |
| Consumption capacity (kWh/day) | 7.62 | 9.60 | 10.91 |
| Charging capacity (kWh/day) | 7.62 | 9.60 | 10.91 |

The flowchart of the Monte Carlo algorithm for predicting the EV charging load is illustrated in Figure 8. For accurate prediction, the simulation calculates the EV charge load at 1-minute intervals.

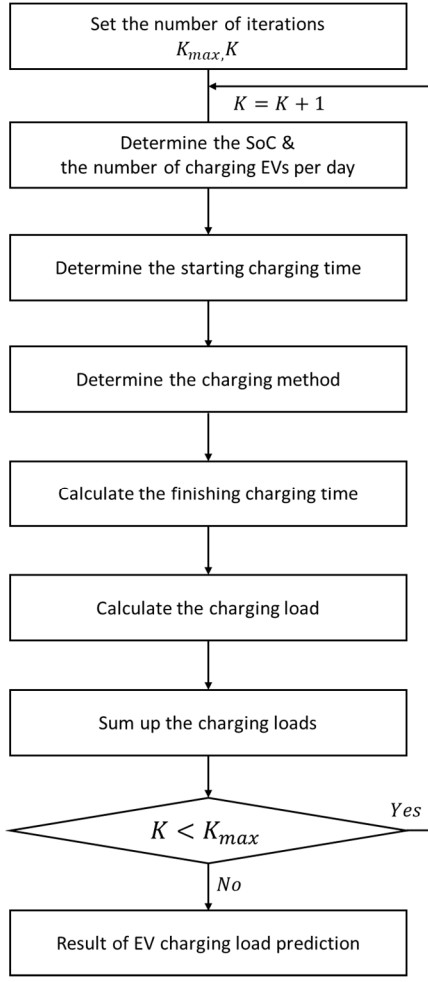

**Figure 8.** The flowchart of the Monte Carlo Simulation for predicting EV charging load curve.

### 3.1. Determining the Parameters for Monte Carlo Simulation

#### 3.1.1. The Number of EVs Charged Per Day and SoC

According to [25], the types of charging behaviors of EV users are divided into "Whenever possible" and "Based on SoC", as shown in Table 13. The users of the "Whenever possible" type, which accounts for 65.1% of all EV users, are assumed to charge the EV unconditionally in their residential area after 6 PM (after work) regardless of the SoC. All EV users drive an average of 48 km per a day and then charge it every day, so the SoC of Ionic, Soul and SM3 are calculated to be 72.79%, 64.44% and 66.94%, respectively, as shown in Table 14. On the other hand, Table 13 shows the users of the "Based on SoC" type, which account for 34.9% of all EV users, charge their EVs based on the SoC regardless of time. The SoC ranges from 20% to 90% with an interval of 10% for each section, and the percentage of EV users starting charging for each section is also shown in Table 13. It is assumed that the detailed SoC values for the "Based on SoC" type of user, which are used in Monte Carlo simulation, are uniformly distributed in each SoC section.

**Table 13.** The charging types of EV users.

| Type | Proportion of EV users | |
|---|---|---|
| Whenever possible (%) | 65.1 | |
| Based on SoC (%) | ≥90    1.78 | |
| | ≥80    4.89 | |
| | ≥70    6.88 | |
| | ≥60    3.35 | 34.9 |
| | ≥50    5.13 | |
| | ≥40    4.22 | |
| | ≥30    4.43 | |
| | ≥20    4.22 | |

**Table 14.** The SoC of "Whenever possible" type users.

| EV Model | Ionic | Soul | SM3 |
|---|---|---|---|
| Battery capacity (kWh) | 28 | 27 | 33 |
| Charging capacity (kWh/Times) | 7.62 | 9.60 | 10.91 |
| The number of charging times (Times/day) | 1 | 1 | 1 |
| SoC (%) | 72.79 | 64.44 | 66.94 |

To predict the charging load curve of EVs by Monte Carlo simulation, it is necessary to determine the number of EVs charged per day and the SoC of each EV. All EVs of the "Whenever possible" type are immediately used in the simulation because all of them are charged daily. However, EVs of the "Based on SoC" type have a different number of times they are charged per day depending on the SoC range as shown in Table 15, so these values are taken into account to determine the number of EVs for Monte Carlo Simulation. The number of charging times per day in each section of SoC in Table 14 is calculated by Equation (1).

$$Number\ of\ charging\ times_{m,k} = \frac{Charging\ capacity_m}{SoC_{m,k}}\ (m = 1,2,3,\ k = 1,2,\ldots,8) \tag{1}$$

where $Number\ of\ charging\ times_{m,k}$ is the number of charging times per day of the $m$ model EV in the k-th section, $Charging\ capacity_m$ is the $m$ model EV's charging capacity per day, and $SoC_{m,k}$ is the SoC of $m$ model EV in the k-th section.

- EV model: $m = 1$ is Ionic, $m = 2$ is Soul, and $m = 3$ is SM3
- k-th section: $k = 1$ is 90%, $k = 2$ is 80%, $\ldots$ , and $k = 8$ is 20%

- It is assumed that the EV user of the "Based on SoC" type charging the EV after 6PM does not charge more than once a day, so if the calculated value of Number of charging times exceeds 1, it is assumed to be 1.

**Table 15.** The number of charging times per day "Based on SoC" type users charge their EVs.

| SoC | ≥90% | ≥80% | ≥70% | ≥60% | ≥50% | ≥40% | ≥30% | ≥20% |
|---|---|---|---|---|---|---|---|---|
| Ionic (Times/day) | 1 | 0.91 | 0.68 | 0.54 | 0.45 | 0.39 | 0.34 | 0.27 |
| Soul (Times/day) | 1 | 1 | 0.89 | 0.71 | 0.59 | 0.51 | 0.44 | 0.36 |
| SM3 (Times/day) | 1 | 1 | 0.83 | 0.66 | 0.55 | 0.47 | 0.41 | 0.33 |

### 3.1.2. Charging Start Time

To determine the charging start time of EVs after work hours, the statistical data in South Korea are used. Table 16 presents data regarding the time that office workers leave, and Table 17 presents the time it takes to arrive home from work. Assuming that the EV users charge their EV as soon as they arrive home, the time when the EV users determined using the data arrive at the home is the charging start time. The charging start time of each EV is also determined stochastically based on the uniform distribution with reference to the proportion of Tables 16 and 17, respectively.

**Table 16.** Data regarding the time when office workers leave the office in South Korea.

| Time | Before 6 PM | 6 PM~7 PM | 7 PM~8 PM | 8 PM~9 PM | After 9 PM |
|---|---|---|---|---|---|
| Rate (%) | 8.0 | 62.9 | 19 | 5.2 | 4.9 |

**Table 17.** Data regarding how long it takes to arrive home from the work in South Korea.

| Time (Min) | Under 30 | 30~60 | 60~90 | 90~120 | 120~150 | 150~180 | Over 180 |
|---|---|---|---|---|---|---|---|
| Rate (%) | 32.7 | 33.9 | 12.5 | 9.6 | 4.0 | 4.6 | 2.7 |

### 3.1.3. Charging Method

Personal chargers in South Korea are divided into rapid chargers and slow chargers. Therefore, it is necessary to consider which chargers are being used to predict the EV charging load curve. According to the report in [25], 86.2% of EV users in South Korea own personal chargers, of which the proportion of regular and fast chargers is shown in Table 18. Information on the charging speed of each charger is also shown in Table 18.

**Table 18.** Information about each charging method.

| | Fast | Regular | |
|---|---|---|---|
| | | Installation | Portable |
| Charging speed (kWh) | 50 | 7 | 3 |
| Ratio (%) | 5.7 | 35.8 | 64.2 |

### 3.1.4. Finishing Charging Time

The above process determines the number of EVs that get charged per day, SoC of the EV, charging start time, and charging method. Finishing charging time is calculated using these parameters. First, the charging capacity of the i-th EV is calculated by Equation (2):

$$t_i = (1 - SoC_i) \times B_i \times \frac{C_i}{60} \quad (i = 1, 2, \ldots, N) \tag{2}$$

where $t_i$ is the time required to charge the i-th EV, N is the number of EVs that get charged per day, $B_i$ is the battery capacity of the i-th EV, and $C_i$ is the charging speed of the charger used for the i-th EV.

The finishing charging time of the i-th EV is calculated using the charging start time and $t_i$ of the i-th EV, expressed in Equation (3) as

$$F_i = S_i + t_i \quad (i = 1, 2, \dots, N) \tag{3}$$

where $F_i$ is the finishing charging time and $S_i$ is the charging start time of the i-th EV.

### 3.1.5. Summing Up the Charging Loads

Equation (4) can be used to calculate the charging load of the i-th EV, and the total charging load can be obtained by summing up all EV charging loads using Equation (5).

$$L_i = \sum_{t=S_i}^{F_i} \frac{C_i}{60} \quad (i = 1, 2, \dots, N) \tag{4}$$

where $L_i$ is the energy requirement for the i-th EV.

$$E_t = \sum_{i=1}^{N} L_i \quad (t = 1, 2, \dots, 1440) \tag{5}$$

where $E_t$ is the total charging load at time $t$.

### 3.1.6. EV Charging Load Curve Based On Monte Carlo Simulation

The annual sales target of EVs, according to the report in [8], are shown in Table 19. The cumulative number of EVs by year was predicted using linear approximation, the results of which are shown in Table 20. In this study, only the number of EVs in South Korea was predicted by excluding the number of EVs in Jeju Island by year, referring to the report in [27]. Figure 9 shows two cases of the predicted EV charging load curve, which are the expected case and the maximum case, from 2017 to 2030 predicted using 5,000 iterative simulations with reference to Table 20.

**Table 19.** The annual sales target of EVs.

| Year | 2019 | 2020 | 2022 | 2025 | 2030 | Cumulative |
| --- | --- | --- | --- | --- | --- | --- |
| **Sales(Thousand)** | 42 | 78 | 153 | 270 | 440 | 3000 |

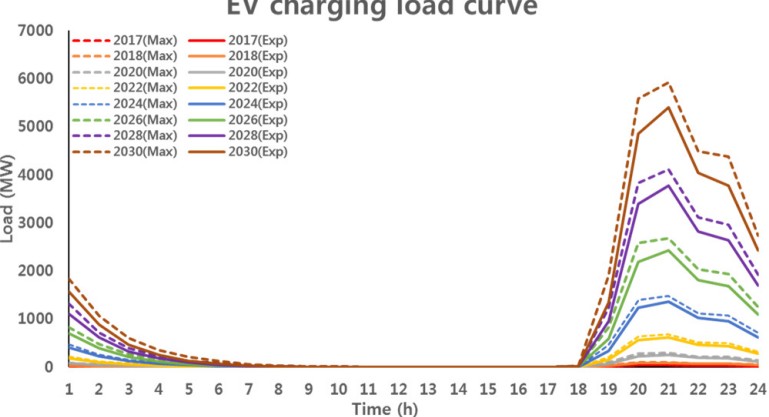

**Figure 9.** The predicted EV charging load curve from 2017 to 2030.

**Table 20.** The predicted number of EV in South Korea excluding Jeju Island by 2030.

| Year | 2017 | 2018 | 2019 | 2020 | 2021 | 2022 | 2023 |
|---|---|---|---|---|---|---|---|
| **Sales(Thousand)** | 16.43 | 40.94 | 70.07 | 121.86 | 198.05 | 298.64 | 463.55 |
| **Year** | 2024 | 2025 | 2026 | 2027 | 2028 | 2029 | 2030 |
| **Sales(Thousand)** | 660.69 | 898.43 | 1171.78 | 1480.74 | 1825.30 | 2205.47 | 2621.57 |

## 4. Changes in the Net Load Curve According to Increased PV and EV Penetration

Changes in the net load curve and peak load time considering the increase in PV penetration are presented in Section 2. This section verifies the changes in the net load curve and peak load time as an additional consideration of the increase in EV penetration. Figure 10 shows net load curve with EV charging load curve added to net load curve considering only PV penetration. When comparing Figure 7 considering only the PV penetration, the duck curve will deteriorate as the EV charging load curve is added after 6 PM on the existing net load curve.

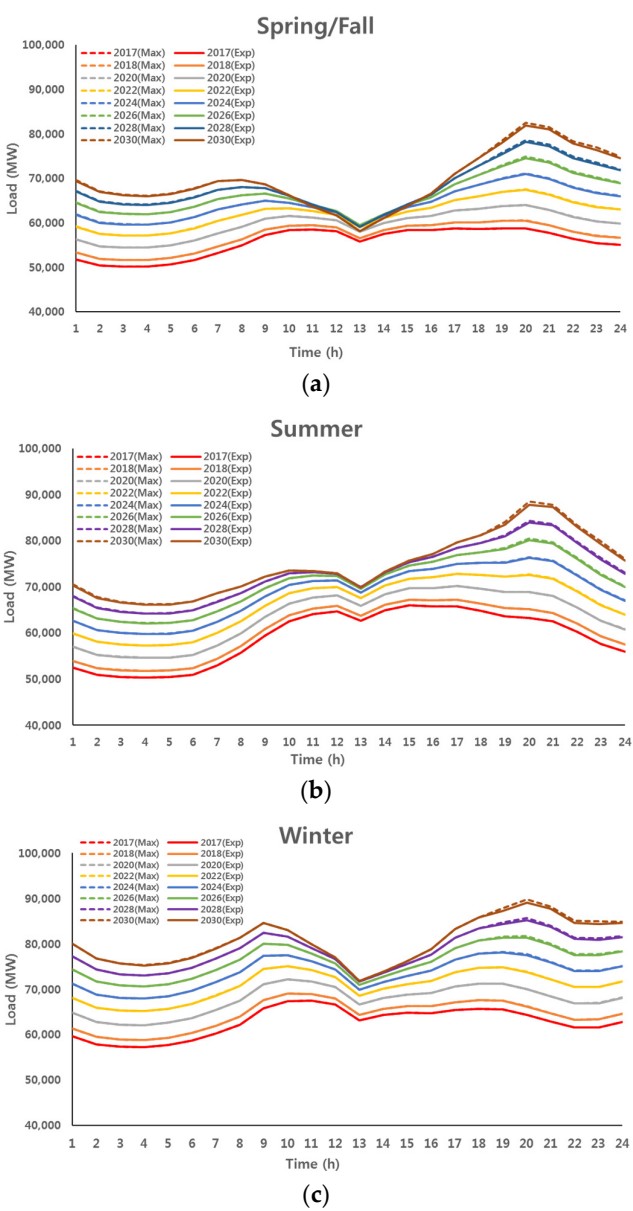

**Figure 10.** Seasonal predicted net load curves in each year from 2017 to 2030 considering PV and EV penetration: (**a**) for spring and fall; (**b**) for summer; and (**c**) for winter.

Table 21 shows annual peak load time considering both PV and EV penetration, and is divided into Expected and Maximum cases according to the EV charging load curve. Compared to Table 11, which shows the peak load time considering only PV penetration, an additional increase in EV penetration accelerates the shift of the seasonal peak load time. The peak load time for spring/fall considering only PV penetration is already 8 PM from 2018, which is not suitable for ToU rate application and DR operation time. Therefore, the results of additional consideration of EV penetration are the same. The peak load time in summer considering only the increase in PV penetration is not suitable for the application of ToU tariff and DR operating time from 2024, but the peak load time considering both PV and EV penetration is not suitable from 2023, which is one year earlier. In winter, the peak load time considering only PV penetration is suitable until 2030 for ToU tariff application and DR operating time. On the other hand, the peak load time considering additional EV penetration is not suitable for the application of ToU tariff and DR operating time from 2027 in the Expected case and 2026 in the Maximum case.

**Table 21.** Peak load time by season from 2018 to 2030 considering PV and EV penetration.

| Year | Expected case | | | Maximum case | | |
|------|-----------|--------|--------|-----------|--------|--------|
|      | Spring/Fall | Summer | Winter | Spring/Fall | Summer | Winter |
| 2018 | 8 PM | 3 PM | 10 AM | 8 PM | 3 PM | 10 AM |
| 2019 | 8 PM | 5 PM | 10 AM | 8 PM | 5 PM | 10 AM |
| 2020 | 8 PM | 5 PM | 10 AM | 8 PM | 5 PM | 10 AM |
| 2021 | 8 PM | 5 PM | 10 AM | 8 PM | 5 PM | 10 AM |
| 2022 | 8 PM | 5 PM | 10 AM | 8 PM | 5 PM | 10 AM |
| 2023 | 8 PM | 8 PM | 10 AM | 8 PM | 8 PM | 10 AM |
| 2024 | 8 PM | 8 PM | 7 PM | 8 PM | 8 PM | 7 PM |
| 2025 | 8 PM | 8 PM | 7 PM | 8 PM | 8 PM | 7 PM |
| 2026 | 8 PM | 8 PM | 7 PM | 8 PM | 8 PM | 8 PM |
| 2027 | 8 PM | 8 PM | 8 PM | 8 PM | 8 PM | 8 PM |
| 2028 | 8 PM | 8 PM | 8 PM | 8 PM | 8 PM | 8 PM |
| 2029 | 8 PM | 8 PM | 8 PM | 8 PM | 8 PM | 8 PM |
| 2030 | 8 PM | 8 PM | 8 PM | 8 PM | 8 PM | 8 PM |

## 5. Conclusions

In this paper, the load curve and net load curve are predicted to confirm the change of the net load curve for each season by year considering high PV and EV penetrations until 2030. PV installed capacity, PV generation, and number of EVs until 2030 are annually estimated by taking South Korea's energy policy into account. In addition, Monte Carlo Simulation is performed considering the number of EVs charged per day, the SoC of each EV, the charging method, the charging start time, and the charging finish time to predict the EV charging load curve after 6 PM, which promotes the duck curve and the shift of peak load time. It also analyzes the change of the net load curve taking PV and EV penetration into account to determine the exact year when ToU tariffs and DR operating times optimized for South Korea's current net load curve are no longer suitable for the season.

When considering only the penetration of PV, the currently operating systems are no longer suitable, since the peak load time is expected to shift to 8 PM in spring/fall of 2018 and in summer of 2025. In addition, when considering EV penetration further, the peak load time shifts to 8 PM in spring/fall of 2018, summer at 2023, and winter of 2027 in the expected case of Monte Carlo Simulation. In the maximum case, the peak load time shifts to 8 PM in spring/fall of 2018, in summer of 2023. In particular, in the maximum case for winter season, the peak load time shifts to 8 PM in 2026, one year earlier than the expected case.

Each future seasonal curve drawn in this paper can be slightly changed due to variability in PV and EV penetration target values. In addition, used data such as load curve, net load curve, PV penetration, and EV penetration are only useful in South Korea. Therefore, the results of this paper

are not suitable for application in other countries. However, the methodology is useful, so if using the same kind of data optimized for each country can get results that are suitable for that country.

The development of DER is essential for improving environmental issues such as GHG reduction. Therefore, it presents the necessity of further research on new policies and institutions considering changes in the net load curve due to DER development.

**Author Contributions:** Conceptualization, C.-Y.K., C.-R.K. and S.-H.C.; methodology, C.-Y.K.; data curation, C.-R.K. and D.-K.K.; writing-original draft preparation, C.-Y.K.; writing-review and editing, C.-Y.K. and S.-H.C.; supervision, S.-H.C. All authors have read and agreed to the published version of the manuscript.

**Funding:** The APC was funded by KETEP (grant number: 2018201060010C).

**Acknowledgments:** This work was supported by the Korea Institute of Energy Technology Evaluation and Planning (KETEP) and the Ministry of Trade, Industry & Energy (MOTIE) of the Republic of Korea (No. 2018201060010C).

**Conflicts of Interest:** The authors declare no conflict of interest.

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
