# Peer review of "Analysis of Challenges Due to Changes in Net Load Curve in South Korea by Integrating DERs"

_electronics, doi:10.3390/electronics9081310_

Round 1

Reviewer 1 Report

PRESENTATION COMMENTS:
Abstract
The abstract does not compare the key results with previously published work.
The abstract does not give the main conclusions or indicate any limitations.
Introduction
There is no review of previous work.
The introduction does not sufficiently review earlier findings.
The introduction does not clearly state what has been done that has not been done before.
The introduction does not indicate the comparators used to demonstrate relevance.
The manuscript discussed the need for the work carried out in Korea. However, a discussion of similar works is missing.
Manuscript Body
The clarity of the presentation can be improved. The title of graphs and figures need to be improved. Most of the graph axis is not properly labelled - the authors tend to use only units for the titles of the axis - mostly kW for the y-axis and hour/minutes for the x-axis
Conclusions
Important results and their consequences are missing.
Any reservations and/or limitations have not been indicated.
The paper does not have a conclusions section (but the last section is titled discussions). Kindly include a conclusions section that will:

(1) bring together the most important results and their consequences

(2) Talk about reservations and limitation
References

The existing references do not indicate alternative techniques which can be used to compare your work.

Formatting
Figure titles and labels need to be reviewed - particularly, the y-axis and x-axis labels for most figures are missing, the authors provide only the units.

TECHNICAL COMMENTS:

The paper is technically sound. However, it is not very clear what the contributions of the paper is/are. The contributions will become clearer if the authors can itemize them within the introduction.

A copy of the paper with some comments is attached.

Author Response

[Authors’ Response]

We greatly appreciate the comments and suggestions from reviewer 1, by which, we think, this research paper is more logically reinforced.

  1. Abstract

- The abstract does not compare the key results with previously published work.

- The abstract does not give the main conclusions or indicate any limitations.

☞ In order to address 2 problems that the reviewer pointed out, we have modified the related part of abstract to “And since ToU (Time-of-Use) tariff and Demand Response (DR) programs are very sensitive to changes in the net load curve, it is very essential to predeict the hourly net load pattern accurately for the modification of pricing and demand response programs in future. However, a long-term demand forecast in South Korea provides only the total amount of annual load (TWh) and the expected peak load level (GW) in summer and winter seasons until 2030.” to present the drawback of the previous national publication and to emphasize our contribution.

Please refer to lines 18 ~ 22 at page 1.

Thank you very sincerely for the good comments.

  1. Introduction

- There is no review of previous work.

☞ As the reviewer pointed out, we have added some sentences and more references like “Previous studies address various problems in power systems such as ramping events, peak load time shifts due to duck curves and irregular generation depending on weather condition [12-13]. And the reference [14] says that the increasing PV penetration ratio and generation can cause a net load demand around the afternoon time to significantly decrease, resulting in an increase in the amount of attenuation of thermal power generators, which in turn leads to economic losses.” and “Other previous work such as 8th Basic Plan for Long-term Electricity Supply and Demand [16] mainly deal with overall negative impacts of the duck curve, and long-term power demand forecasting including the total amount of annual load (TWh) and the expected peak load level (GW) in summer and winter seasons until 2030. However, in order to accurately analyze the change of net load curve pattern, it is necessary to predict the hourly pure load curve and the hourly net load curve which includes the pure load level, demand response, renewable generation and other factors. Some studies introduced the hourly net load pattern with renewable energy resources, but they were based on only the current load level and mainly focused on the impact of variability of renewables [12, 17]. Now, we need for the hourly load pattern prediction considering the future change of energy circumstances until 2030.” to review previous studies and to emphasize our contribution in the introduction section.

Please refer to lines 60 ~ 65, and lines 73 ~ 82 at page 2.

Thank you very sincerely for the good comments.

- The introduction does not sufficiently review earlier findings.

☞ As the reviewer pointed out, we have modified the related part of introduction to “Figure 1 shows a difference of actual net load curves in South Korea on Nov. 20 and Nov. 21 in 2018. According to the Korea Meteorological Administration (KMA) information, it was clear on Nov. 20 and cloudy on Nov. 21 [15]. Total amounts of energy consumption were 1,525,173 GWh on Nov. 20 and 1,557,242 GWh on Nov. 21, which means that the difference of energy consumption was 32,069 GWh in just one day because of the change of weather condition. When we consider the additional EV charging effect, it will make the duck curve worse, especially after 8 PM when the need for EV charging is intensive, by making a large difference between the peak load and the off-peak load.” to provide more reviews of the Reference 14 as shown in Figure 1.

Please refer to lines 65 ~ 72 at page 2.

Thank you very sincerely for the good comments.

- The introduction does not clearly state what has been done that has not been done before.

☞ As the reviewer pointed out, we have added a paragraph of “However, in order to accurately analyze the change of net load curve pattern, it is necessary to predict the hourly pure load curve and the hourly net load curve which includes the pure load level, demand response, renewable generation and other factors. Some studies introduced the hourly net load pattern with renewable energy resources, but they were based on only the current load level and mainly focused on the impact of variability of renewables [12, 17]. Now, we need for the hourly load pattern prediction considering the future change of energy circumstances until 2030. Thus, we predict the hourly load curve and the hourly net load by taking into account PV installed capacity, PV generation, and the number of EVs based on the target of Korean policy until 2030.” to clarify what we do in our study.

Please refer to lines 76 ~ 84 at page 3.

Thank you very sincerely for the good comments.

- The introduction does not indicate the comparators used to demonstrate relevance.

☞ Our paper is written to analyze the changes of the future net load curve by predicting the hourly load curve considering the future PV and EV penetration changes until 2030, which has not been previously performed. We have tried to find other similar study cases in both Korea and international works. But we failed to find the related comparator for verification.

Thank you very sincerely for the good comments.

- The manuscript discussed the need for the work carried out in Korea. However, a discussion of similar works is missing.

☞ As the reviewer pointed out, we have added some paragraphs of “Previous studies address various problems in power systems such as ramping events, peak load time shifts due to duck curves and irregular generation depending on weather condition [12-13]. ~ When we consider the additional EV charging effect, it will make the duck curve worse, especially after 8 PM when the need for EV charging is intensive, by making a large difference between the peak load and the off-peak load.” and “Other previous work such as 8th Basic Plan for Long-term Electricity Supply and Demand [16] mainly deal with overall negative impacts of the duck curve, and long-term power demand forecasting including the total amount of annual load (TWh) and the expected peak load level (GW) in summer and winter seasons until 2030. ~ Thus, we predict the hourly load curve and the hourly net load by taking PV installed capacity into account, PV generation, and the number of EVs based on the target of Korean policy until 2030.” to discuss the similar works and the necessity of our study.

Please refer to lines 60 ~ 84 at page 2.

Thank you very sincerely for the good comments.

  1. Manuscript Body

- The clarity of the presentation can be improved. The title of graphs and figures need to be improved. Most of the graph axis is not properly labelled - the authors tend to use only units for the titles of the axis - mostly kW for the y-axis and hour/minutes for the x-axis

☞ As the reviewer pointed out, we have modified the titles of the graphs and figures to match the MDPI format. Also, we have modified the titles of the axis appropriately.

Thank you very sincerely for the good comments.

  1. Conclusions

Important results and their consequences are missing.

Any reservations and/or limitations have not been indicated.

The paper does not have a conclusions section (but the last section is titled discussions). Kindly include a conclusions section that will:

(1) bring together the most important results and their consequences

(2) Talk about reservations and limitation

☞ As the reviewer pointed out, we have modified the chapter 5. Conclusion as a whole, adding main contribution, limitation, application and future work.

Please refer to lines 366 ~ 390 in page 17.

Thank you very sincerely for the good comments.

  1. References

- The existing references do not indicate alternative techniques which can be used to compare your work.

☞ As the reviewer pointed out, we have added more references in the paper to supplement the weakness and to compare with our work directly. We have tried to find other similar study cases in both Korea and international works. But we failed to find the related comparator, as we mentioned before.

Thank you very sincerely for the good comments.

  1. Formatting

- Figure titles and labels need to be reviewed - particularly, the y-axis and x-axis labels for most figures are missing, the authors provide only the units.

 â˜ž As the reviewer pointed out, we have modified the titles of the graphs and figures to match the MDPI format. Also, we have added y-axis and x-axis labels of figures

Thank you very sincerely for the good comments.

Reviewer 2 Report

The PV generation and EV penetration are considered in this paper to predict the future net load curve in South Korea. It is concluded that the existing application of ToU tariff and DR operating time is no longer appropriate. The subject of high interesting for optimized energy use and supply.

However, it is lack of literature review on the subject and hard to assess the contribution or novelty of this paper. Please revise it to have previous works on the subject and make full discussion on methodologies used and results obtained in association with any progress made previously.

Some specific problems found:

  • The keywords need to be reconfirmed. For example, the paper revolves around PV generation and EV penetration, therefore the keyword PV generation is even more important than SoC (State-of-Charge).
  • In line 85, the ‘2030’ should be ‘2017’.
  • In line 90, the ‘15:00’ should be ‘17:00’ according to the Table 2.
  • In line 171, the additional ‘(1)’ should be removed.
  • Some format errors in the manuscript such as Figure 5, Figure 6, Figure 7, and Figure 8 should be formatted carefully.

A large number of formatting problems are found in the titles of figures and tables and the line styles of tables such as Table 14, Table 18 and Table 19, all of the errors must be formatted and revised carefully according to the MDPI article template

Author Response

[Authors’ Response]

We greatly appreciate the comments and suggestions from reviewer 2, by which, we think, this research paper is more logically reinforced.

  1. However, it is lack of literature review on the subject and hard to assess the contribution or novelty of this paper. Please revise it to have previous works on the subject and make full discussion on methodologies used and results obtained in association with any progress made previously.

☞ As the reviewer pointed out, we have added some paragraphs and more references of “Previous studies address various problems in power systems such as ramping events, peak load time shifts due to duck curves and irregular generation depending on weather condition [12-13]. And the reference [14] says that the increasing PV penetration ratio and generation can cause a net load demand around the afternoon time to significantly decrease, resulting in an increase in the amount of attenuation of thermal power generators, which in turn leads to economic losses.” and “Other previous work such as 8th Basic Plan for Long-term Electricity Supply and Demand [16] mainly deal with overall negative impacts of the duck curve, and long-term power demand forecasting including the total amount of annual load (TWh) and the expected peak load level (GW) in summer and winter seasons until 2030. However, in order to accurately analyze the change of net load curve pattern, it is necessary to predict the hourly pure load curve and the hourly net load curve which includes the pure load level, demand response, renewable generation and other factors. Some studies introduced the hourly net load pattern with renewable energy resources, but they were based on only the current load level and mainly focused on the impact of variability of renewables [12, 17]. Now, we need for the hourly load pattern prediction considering the future change of energy circumstances until 2030.” to introduce the previous studies and compare them with our study.

 Please refer to lines 60 ~ 65, and lines 73 ~ 82 at page 2.

Thank you very sincerely for the good comments.

  1. The keywords need to be reconfirmed. For example, the paper revolves around PV generation and EV penetration, therefore the keyword PV generation is even more important than SoC (State-of-Charge).

☞ [Line 30] As the reviewer pointed out, we have removed SoC (State-of-Charging) and have added PV generation in keywords part.

Thank you very sincerely for the good comments.

  1. In line 85, the ‘2030’ should be ‘2017’.

☞ As the reviewer pointed out, we have changed ‘2030’ into ‘2017’.

Thank you very sincerely for the good comments.

  1. In line 90, the ‘15:00’ should be ‘17:00’ according to the Table 2.

☞ As the reviewer pointed out, we have changed the time format for ’17:00’ into ‘5 PM’. And all of the time has been changed into AM or PM form.

Thank you very sincerely for the good comments.

  1. In line 171, the additional ‘(1)’ should be removed.

☞ As the reviewer pointed out, we have removed ‘(1)’ from the text.

Thank you very sincerely for the good comments.

  1. Some format errors in the manuscript such as Figure 5, Figure 6, Figure 7, and Figure 8 should be formatted carefully.

☞ As the reviewer pointed out, we have modified the format of all the Figures in the paper according to the MDPI format.

Thank you very sincerely for the good comments.

Reviewer 3 Report

The topic of this work is interesting. Some extended works need to be done before the manuscript is accepted. Below are some detailed issues with this manuscript:

  1. On page 2, the authors claimed that Figure 1 indicates an additional power generation of 32,069GWh was needed in just one day. How did the authors obtain this number? Some explanations are needed.
  2. A reference is needed for Table 1 and Table 2.
  3. The authors Figure 2 is based on the actual data obtained from KPX. A link or at least a reference is needed.
  4. The authors wrote “However, the peak load time for the spring/fall changed from 11:00 in 2015 and 2016 to 15:00 in 2017, …” at the end of page 3. Table 2 indicates a different result. Is this a typo? Please check the whole manuscript. This kind of mistake affects the quality of the work. Also, it is not clear why the peak load time doesn’t shift in the summer and winter seasons.
  5. In section 2.1, it is not clear which data is measured by the authors and which ones are from the references. Also, in this section, the authors mentioned three curves (load curve, net load curve, and PV generation curve), and the reviewer only saw the discussion of two figures. In addition, three years (2016, 2017, and 2018) were mentioned, which year’s data is known, and which year’s data needs to be predicted? Please make it clear.
  6. The authors chose Monte Carlo simulation. Since there are so many other methods, why did the authors choose this method? Some explanations are needed.
  7. This study focuses on the analysis of the net load curve in South Korea. Can the results of this study be applied to other countries around the world? A discussion is needed since this can be useful.

Author Response

[Authors’ Response]

We greatly appreciate the comments and suggestions from reviewer 3, by which, we think, this research paper is more logically reinforced.

  1. On page 2, the authors claimed that Figure 1 indicates an additional power generation of 32,069GWh was needed in just one day. How did the authors obtain this number? Some explanations are needed.

☞ As the reviewer pointed out, we have added information on the power generation at 20/11/2018 and 21/11/2018 respectively, and the difference of power generation between 20/11/2018 and 21/11/2018 in Table 1.

Table 1. Net load difference between 20/11/2018 and 21/11/2018

20/11/2018

21/11/2018

Net load (GWh)

1,525,173

1,557,242

Difference (GWh)

32,069

Thank you very sincerely for the good comments.

  1. A reference is needed for Table 1 and Table 2.

☞ As the reviewer pointed out, we have added the reference [18] to Table 2 (Table 1 in the original version). And Table 3 (Table 2 in the original version) has no reference since it represents the contents of Figure 2.

Thank you very sincerely for the good comments.

  1. The authors Figure 2 is based on the actual data obtained from KPX. A link or at least a reference is needed

☞ As the reviewer pointed out, we have added an internet link of KPX in the reference [20].

Thank you very sincerely for the good comments.

  1. The authors wrote “However, the peak load time for the spring/fall changed from 11:00 in 2015 and 2016 to 15:00 in 2017, …” at the end of page 3. Table 2 indicates a different result. Is this a typo? Please check the whole manuscript. This kind of mistake affects the quality of the work. Also, it is not clear why the peak load time doesn’t shift in the summer and winter seasons.

☞ As the reviewer pointed out, we have checked all time forms and changed 15:00 to 5 PM, matching with Table values. Also, we have added a sentence of “This fact means that cooling and heating loads are not frequently used in spring and fall unlike summer and winter, so the peak load time due to increasing PV generation is shifted earlier than other seasons.”.

Please refer to lines 117 ~ 119 at page 4.

Thank you very sincerely for the good comments.

  1. In section 2.1, it is not clear which data is measured by the authors and which ones are from the references. Also, in this section, the authors mentioned three curves (load curve, net load curve, and PV generation curve), and the reviewer only saw the discussion of two figures. In addition, three years (2016, 2017, and 2018) were mentioned, which year’s data is known, and which year’s data needs to be predicted? Please make it clear.

☞ As the reviewer pointed out, we have modified the part as a whole to minimize the reader’s confusion. Please check the chapter of 2.1.

Thank you very sincerely for the good comments.

6 The authors chose Monte Carlo simulation. Since there are so many other methods, why did the authors choose this method? Some explanations are needed.

☞ As the reviewer pointed out, we have added some sentences of “The data used to predict EV charging load is probability-based data based on surveys of actual EV users. Monte Carlo Simulation is the most suitable method to utilize probability-based data as a method of performing a simulation by repeating random selection of input variables from a probability distribution [23-24].” to explain why we used Monte Carlo simulation technique.

Please refer to lines 233 ~ 236 at page 10.

Thank you very sincerely for the good comments.

  1. A large number of formatting problems are found in the titles of figures and tables and the line styles of tables such as Table 14, Table 18 and Table 19, all of the errors must be formatted and revised carefully according to the MDPI article template.

☞ As the reviewer pointed out, we have modified the format of all the Figures in the paper according to the MDPI format.

Thank you very sincerely for the good comments.

Reviewer 4 Report

This work discusses the concerns over the changes of the net load due to DER integration into the power system, which may lead to a negative impact on the existing operation of the power systems in South Korea.  The topic of the paper is interesting. However, the reviewer has some comments on this study, which need authors to address as follows,

1) Figure 1 shows the load curves over the two-day period. According to the diagrams, the authors point out that the discrepancy between the two days is due to the variation of the PV generation. However, there are neither obvious evidence nor references in the paper to support the authors’ comments.

2) The authors has drawn the conclusion (from lines 65 to 71) that both of the existing TOU and DR are improper to the situation caused by the changes of PV generation and EV consumption. However, there is no evidence or data to support the conclusion directly  from the beginning of the manuscript to the content there.

3)  Lines 89 and 90 present “…… the peak load time for the spring/fall changed from 11:00 in 2015 and 2016 to 15:00 in 2017……”. However, Table 2 presents the peak in Spring/Fall, 2017 at 17:00. Moreover, there are limited differences between the 11am, 5pm and 8pm in Spring/Fall, 2017 in Figure 2(a). The authors need to clarify where exactly is the peak in the authors’ view. The accurate data should be shown to support the conclusion drawn by the authors.

4) The author has cited the report “The 8th Basic Plan for Long-term Electricity Supply and Demand (2017 - 2031)” for the calculation as presented from lines 104 to 107. This report is made by Ministry of Trade, industry and Energy, South Korea and demonstrates the actual data for both accumulated consumption and peak demand in 2016 and 2017 (page 70). However, the data in this report (over 80GW) has relatively large discrepancy to the data in Figure 2 (below 70 GW) in the manuscript. The author need to explain where the data is adopted from. A reference is needed to further explain the issue.   

5) From line 101 to line 113, the authors uses 4 steps to predict the load curves. In the last step, the author uses the load curves in 2018 to minus the PV generation. However, Neither theoretical or practical methods, evidences or references to support correctness of this calculation. Convincing evidences are needed if the calculation is believed correct.

5) Table 3 provides a series of numbers in line with the load growth rates as stated by the author from Ref. 15. However, the numbers on page 74, ref. 15 shows apparently difference from the data in Table 3 in the manuscript. For instance, the rate in 2025 in the paper is 1.94% while being 0.6 on page 74 in Ref. 15. The author need to explain which parts of the data are cited from the reference or how the data in the table is received.

6) In lines 206-208, the authors state the numbers in Table 12 are derived on the basis of the principle of uniform distribution. However, these numbers are apparently incompatible to this principle. The authors need to explain how these data are derived. Or some references associated with the calculation methods should be added.

7)  Table 14 provides the accurate numbers to estimate the charging numbers for different EV users. The authors need to provide the evidence or calculation methods how the numbers are defined since there are no explanations on this issue.

8) The authors has drawn the conclusion that the ToU and DR scheme will be improper to operating the future power systems due to the load curves changed. However, there are lack of the necessary details to support the conclusions.

9) Both of the limitations and the future work are necessary in the conclusion or introduction.

10) Conclusions are needed to be improved.

Author Response

[Authors’ Response]

We greatly appreciate the comments and suggestions from reviewer 4, by which, we think, this research paper is more logically reinforced.

  1. Figure 1 shows the load curves over the two-day period. According to the diagrams, the authors point out that the discrepancy between the two days is due to the variation of the PV generation. However, there are neither obvious evidence nor references in the paper to support the authors’ comments.

☞ 20/11/2018 is Tuesday and 21/11/2018 is Wednesday. Since the two days are weekdays, not holidays, they should have a similar hourly pattern of net load curve. However, the amount of power generation by PV has a large difference due to the weather conditions during daylight hours. The below table shows the actual cloud information of 20/11/2018 and 21/11/2018 obtained from Korea Meteorological Administration. In the table, the 0 to 5 levels mean clear, the 6 to 8 level mean cloudy, and the 9 to 10 levels mean heavily cloudy. As the reviewer pointed out, we have added a sentence of “According to the Korea Meteorological Administration (KMA) information, it was clear on Nov. 20 and cloudy on Nov. 21 [15].” and its reference.

Please refer to lines 66 ~ 67 at page 2.

Thank you very sincerely for the good comments.

Table. Weather conditions on 20/11/2018 and 21/11/2018 in South Korea.

Hour

20/11/2018

21/11/2018

7

0

10

8

0

10

9

0

10

10

0

10

11

0

10

12

0

10

13

0

10

14

0

10

15

0

10

16

0

10

17

1

10

18

2

10

  1. The authors has drawn the conclusion (from lines 65 to 71) that both of the existing TOU and DR are improper to the situation caused by the changes of PV generation and EV consumption. However, there is no evidence or data to support the conclusion directly from the beginning of the manuscript to the content there.

☞ As the reviewer pointed out, we have modified the related part into “Table 2 shows the current seasonal time range for ToU tariff in Korea [18]. The peak load time will be shifted into other time ranges as shown in Figure 1. Additionally, DR is implemented from 9 AM to 8 PM (except 12 PM ~ 1 PM) to reduce electricity demand in South Korea [19]. Thus, as the number of EVs charging after 8 PM increases, it is necessary to change the DR operating time to manage new peak loads occurring after 8 PM due to EV charging load.” in the introduction and have added an additional explanation of “When considering only the penetration of PV, the currently operating systems are no longer suitable since the peak load time is expected to shift to 8 PM in spring/fall of 2018 and in summer of 2025. And when considering EV penetration further, the peak load time shifts to 8 PM in spring/fall of 2018, summer at 2023, and winter of 2027 in the expected case of Monte-Carlo Simulation. In the maximum case, the peak load time shifts to 8 PM in spring/fall of 2018, in summer of 2023. In particular, in the maximum case for winter season, the peak load time shifts to 8 PM in 2026, one year earlier than the expected case.” in the conclusion.

Please refer to lines 86 ~ 91 at page 3 and lines 376~382 at page 17.

Thank you very sincerely for the good comments.

  1. Lines 89 and 90 present “…… the peak load time for the spring/fall changed from 11:00 in 2015 and 2016 to 15:00 in 2017……”. However, Table 2 presents the peak in Spring/Fall, 2017 at 17:00. Moreover, there are limited differences between the 11am, 5pm and 8pm in Spring/Fall, 2017 in Figure 2(a). The authors need to clarify where exactly is the peak in the authors’ view. The accurate data should be shown to support the conclusion drawn by the authors.

☞ As the reviewer pointed out, we have checked all time forms and changed 15:00 to 5 PM, matching with Table values. Please refer to line 115 at page 4.

The below table shows the data used to draw Figure 2. The seasonal peak load time for each year are indicated in yellow, and the results are shown in Table 2.

Thank you very sincerely for the good comments.

Table. The data used to make Figure 2

Hour

Spring/Fall

(GWh)

Summer

(GWh)

Winter

(GWh)

2015

2016

2017

2015

2016

2017

2015

2016

2017

1

50,000

50,965

51,760

48,354

50,092

51,560

57,123

57,767

59,640

2

48,681

49,565

50,360

47,268

48,476

50,039

55,676

56,036

57,801

3

48,281

49,313

50,147

46,944

48,138

49,641

55,123

55,556

57,280

4

48,240

49,274

50,160

46,757

48,018

49,456

54,869

55,475

57,199

5

48,572

49,573

50,608

46,846

48,095

49,539

55,165

55,823

57,683

6

49,119

50,302

51,606

47,093

48,389

50,097

55,694

56,515

58,661

7

50,248

51,715

53,178

48,188

49,778

52,029

56,499

57,658

60,260

8

51,838

53,498

54,859

50,196

52,256

54,759

57,829

59,351

62,160

9

54,753

56,349

57,183

54,215

56,447

58,435

61,565

62,910

65,802

10

56,297

57,804

58,291

57,253

59,816

61,458

63,209

64,455

67,357

11

56,571

58,124

58,500

58,689

61,430

62,998

63,287

64,618

67,431

12

56,332

57,845

58,156

59,314

62,118

63,587

62,646

63,948

66,612

13

53,623

55,122

55,789

57,251

60,154

61,608

59,265

60,520

63,131

14

55,519

56,958

57,481

59,463

62,427

63,877

60,681

61,967

64,390

15

56,339

57,791

58,319

60,427

63,383

64,861

61,126

62,408

64,875

16

56,158

57,672

58,292

60,121

63,107

64,632

60,765

62,158

64,739

17

56,390

57,972

58,746

60,036

63,071

64,630

61,151

62,675

65,423

18

56,122

57,682

58,565

59,072

62,053

63,678

61,349

62,949

65,679

19

56,344

57,823

58,722

57,949

60,860

62,544

61,543

62,953

65,554

20

56,526

57,893

58,722

57,801

60,553

62,193

60,465

61,792

64,284

21

55,713

56,930

57,704

57,370

59,947

61,432

59,293

60,450

62,801

22

54,557

55,619

56,348

55,375

57,758

59,216

58,369

59,329

61,479

23

53,848

54,779

55,412

53,019

55,216

56,621

58,792

59,574

61,507

24

53,091

54,262

55,035

51,414

53,527

54,967

59,852

60,738

62,695

  1. The author has cited the report “The 8th Basic Plan for Long-term Electricity Supply and Demand (2017 - 2031)” for the calculation as presented from lines 104 to 107. This report is made by Ministry of Trade, industry and Energy, South Korea and demonstrates the actual data for both accumulated consumption and peak demand in 2016 and 2017 (page 70). However, the data in this report (over 80GW) has relatively large discrepancy to the data in Figure 2 (below 70 GW) in the manuscript. The author need to explain where the data is adopted from. A reference is needed to further explain the issue.

☞ Figure 2 shows the seasonal average net load curves in South Korea, not peak load curves. The below table shows the data including summer peak demand and winter peak demand for 2016 and 2017 used in Figure 2. Since the values mentioned at page 70 of “The 8th Long-Term Power Supply Basic Plan (2017-2031)” are the peak demand in 2016 and 2017, there is a difference between the two data.

As the reviewer pointed out, we have modified “Figure 2 is the net load curve for South Korea” with “Figure 2 is the seasonal average net load curve for South Korea” to minimize the reader’s confusion.

  Please refer to lines 103 ~ 104 at page 3.

Thank you very sincerely for the good comments.

Table. data including summer peak demand and winter peak demand for 2016 and 2017.

Hour

Sumer

Winter

12/8/2016

21/7/2017

21/1/2016

12/12/2017

1

  63,412.24

  63,866.68

69,490.63

  69,769.53

2

  60,362.54

  61,470.62

67,852.53

  67,961.33

3

  60,140.92

  60,697.16

67,400.88

  67,270.88

4

  60,064.72

  59,855.93

66,876.68

  66,999.54

5

  59,816.66

  59,778.54

66,610.40

  67,241.29

6

  60,327.88

  60,696.72

66,804.30

  68,694.08

7

  62,327.18

  64,404.62

67,916.09

  71,283.56

8

  66,276.63

  69,332.44

70,951.90

  75,410.58

9

  73,238.05

  75,062.07

77,689.93

  82,226.32

10

  79,057.24

  80,372.68

81,633.71

  84,967.60

11

  81,817.23

  82,543.58

82,795.57

  84,883.76

12

  83,010.09

  83,896.51

81,988.53

  83,882.97

13

  79,991.59

  81,012.44

76,348.74

  78,761.58

14

  83,244.03

  83,308.95

78,260.85

  80,830.47

15

  84,939.46

  84,189.38

78,819.44

  82,099.71

16

  84,852.70

  84,110.73

78,059.97

  82,685.55

17

  85,184.64

  84,502.07

78,668.79

  84,490.20

18

  84,000.08

  83,581.88

78,667.71

  84,808.90

19

  81,302.33

  82,152.83

78,183.24

  82,585.33

20

  80,111.93

  81,168.98

75,863.50

  79,562.27

21

  78,488.15

  80,245.88

73,206.43

  76,522.00

22

  74,774.60

  76,982.03

71,300.20

  73,948.14

23

  71,083.99

  73,034.18

71,057.34

  73,189.73

24

  68,565.23

  70,269.56

72,827.82

  74,157.23

  1. From line 101 to line 113, the authors uses 4 steps to predict the load curves. In the last step, the author uses the load curves in 2018 to minus the PV generation. However, Neither theoretical or practical methods, evidences or references to support correctness of this calculation. Convincing evidences are needed if the calculation is believed correct.

☞ As the reviewer pointed out, we have added a sentence and the reference like “A net load curve(or pattern) in a day is obtained by removing a PV generation curve from a load curve [21].” in line 195 and have modified the part as a whole to minimize the reader’s confusion.

Please check the chapter of 2.1.

Thank you very sincerely for the good comments.

  1. Table 3 provides a series of numbers in line with the load growth rates as stated by the author from Ref. 15. However, the numbers on page 74, ref. 15 shows apparently difference from the data in Table 3 in the manuscript. For instance, the rate in 2025 in the paper is 1.94% while being 0.6 on page 74 in Ref. 15. The author need to explain which parts of the data are cited from the reference or how the data in the table is received.

☞ The numbers on page 74 of reference 16 (reference 15 in the original version) are the target demand growth rates, not the load growth rates. The definition of target demand is explained on page 31 of reference 16. In this paper, we used the load growth rates mentioned on page 70 of reference 16, not the target demand growth rates.

Thank you very sincerely for the good comments.

  1. In lines 206-208, the authors state the numbers in Table 12 are derived on the basis of the principle of uniform distribution. However, these numbers are apparently incompatible to this principle. The authors need to explain how these data are derived. Or some references associated with the calculation methods should be added.

☞ As the reviewer pointed out, we have revised the part into “The SoC ranges from 20 % to 90 % with an interval of 10 % for each section, and the percentage of EV users starting charging for each section is also shown in Table 13. It is assumed that the detailed SoC values for the "Based on SoC" type of user, which are used in Monte Carlo simulation, are uniformly distributed in each SoC section.” to minimize the reader’s confusion.

  Please refer to lines 268 ~ 271 at page 12.

Thank you very sincerely for the good comments.

  1. Table 14 provides the accurate numbers to estimate the charging numbers for different EV users. The authors need to provide the evidence or calculation methods how the numbers are defined since there are no explanations on this issue.

☞ As the reviewer pointed out, we have added the equation (1) and its explanation in details to show how the numbers are calculated.

Please refer to lines 277 ~ 288 at page 12.

Thank you very sincerely for the good comments.

  1. The authors has drawn the conclusion that the ToU and DR scheme will be improper to operating the future power systems due to the load curves changed. However, there are lack of the necessary details to support the conclusions

☞ As the reviewer pointed out, we have revised the part of conclusion with “In this paper, the load curve and net load curve are predicted to confirm the change of the net load curve for each season by year considering high PV and EV penetrations until 2030. ~ In particular, in the maximum case for winter season, the peak load time shifts to 8 PM in 2026, one year earlier than the expected case.” to show why they are not suitable for the future seasonal load data.

Please refer to lines 367 ~ 382 at page 17.

Thank you very sincerely for the good comments.

  1. Both of the limitations and the future work are necessary in the conclusion or introduction.

☞ As the reviewer pointed out, we have modified the chapter 5. Conclusion as a whole, adding main contribution, limitation, application and future work.

Please refer to lines 366 ~ 390 in page 17.

Thank you very sincerely for the good comments.

  1. Conclusions are needed to be improved.

☞ As the reviewer pointed out, we have modified the chapter 5. Conclusion as a whole, adding main contribution, limitation, application and future work.

Please refer to lines 366 ~ 390 in page 17.

Thank you very sincerely for the good comments.

Reviewer 5 Report

The article is well written and informative. The following suggestions are made:

  1. The units followed by the values need a space throughout the article.
  2. Fig 1: This is a load curve based on time and not the weather.
  3. Graphs throughout the paper needs re-work in terms of labeling, font size, line width etc.
  4. There are inconsistencies in font types in table 1.
  5. The data are missing in table 11.
  6. Where is Fig. 7?
  7. Factors affecting the simulation results need to be discussed and listed.

Author Response

[Authors’ Response]

We greatly appreciate the comments and suggestions from reviewer 5, by which, we think, this research paper is more logically reinforced.

  1. The units followed by the values need a space throughout the article.

☞ As the reviewer pointed out, we have added a space between number and unit throughout the paper.

Thank you very sincerely for the good comments.

  1. Fig 1: This is a load curve based on time and not the weather.

☞ As the reviewer pointed out, we have changed the title of Figure 1 to "Net load curves for 20/11/2018 and 21/11/2018".

Thank you very sincerely for the good comments.

  1. Graphs throughout the paper needs re-work in terms of labeling, font size, line width etc.

☞ As the reviewer pointed out, we have revised all graphs and tables throughout the paper.

Thank you very sincerely for the good comments.

  1. There are inconsistencies in font types in table 1.

☞ As the reviewer pointed out, we have revised the font type of Table 1 and throughout the paper according to the MDPI format.

Thank you very sincerely for the good comments.

  1. The data are missing in table 11.

☞ As the reviewer pointed out, we have revised the cell and the data of Table 12(Table 11 in the original version.

Please refer to line 219 at page 10.

Thank you very sincerely for the good comments.

  1. Where is Fig. 7

☞ As the reviewer pointed out, we have edited Figure 7 again and rearranged it in page 9.

Please refer to line 214 at page 7.

Thank you very sincerely for the good comments.

  1. Factors affecting the simulation results need to be discussed and listed

☞ As the reviewer pointed out, we have added a paragraph of “Each future seasonal curve drawn in this paper can be slightly changed due to variability in PV and EV penetration target values. In addition, used data such as load curve, net load curve, PV penetration, and EV penetration are only useful in South Korea. Therefore, the results of this paper are not suitable for application in other countries. However, the methodology is useful, so if using the same kind of data optimized for each country can get results that are suitable for that country.” in the conclusion to explain the factors affecting the simulation.

Please refer to lines 383 ~ 387 at page 17.

Thank you very sincerely for the good comments.

Round 2

Reviewer 2 Report

The revisions are acceptable.

Author Response

Thank you sincerely again for your good review comments.

Reviewer 3 Report

The reviewer is okay with the authors' responses. 

Author Response

(The authors gave the same response as above.)

Reviewer 4 Report

The manuscript has been improved. However, there are some concerns left.
1) As Figure 1 shown, the authors point out that the discrepancy between the two days is due to the variation of the PV generation. However, why wind generation or other renewable resources have not been considered except PV in the discrepancy between the two days.
2) The last sentence between line 107 and 108 remains confusion since the authors state that Figure 2 shows the average value of these two years.
3) Grammar and punctuations should be checked throughout the revised manuscript.

Author Response

Thank you sincerely again for your good review comments.

Please check the attached response to reviewer's comments.
